# Fermented *Morinda citrifolia* (Noni) Alleviates DNCB-Induced Atopic Dermatitis in NC/Nga Mice through Modulating Immune Balance and Skin Barrier Function

**DOI:** 10.3390/nu12010249

**Published:** 2020-01-18

**Authors:** Sung Ho Kim, Geum Su Seong, Se Young Choung

**Affiliations:** 1Department of Life and Nanopharmaceutical Sciences, Graduate School, Kyung Hee University, 26 Kyungheedae-ro, Dongdaemun-gu, Seoul 02447, Korea; sevenksh91@gmail.com; 2Korea Food Research Institute, 245 Nongsaengmyeong-ro, Iseo-myeon, Wanju_Gun, Jeollabuk-do 55365, Korea; gsseong824@nate.com; 3Department of Preventive Pharmacy and Toxicology, College of Pharmacy, Kyung Hee University, 26 Kyungheedae-ro, Dongdaemun-gu, Seoul 02447, Korea

**Keywords:** atopic dermatitis, fermented *Morinda citrifolia*, NC/Nga, Th1, Th2, immune balance, skin barrier function

## Abstract

*Morinda citrifolia*, a fruit generally known as “Noni”, has been traditionally used in parts of East Asia to relieve inflammatory diseases. Although several studies using noni have been reported, the effect of fermented *Morinda citrifolia* (F.NONI) on atopic dermatitis (AD) has not been investigated. Thus, we aimed to investigate the improving effect of F.NONI treatment on AD-like skin lesions and elucidate molecular mechanisms. F.NONI was prepared by the fermentation of noni fruit with probiotics and then extracted. F.NONI was orally administrated to NC/Nga mice to evaluate its therapeutic effect on 2,4-dinitrochlorobenzene (DNCB)-induced AD. Oral administration of F.NONI significantly alleviated AD lesions and symptoms such as dermatitis scores, ear thickness, scratching behavior, epidermal thickness, and infiltration of inflammatory cells (e.g., mast cells and eosinophils). In addition, F.NONI treatment reduced the levels of histamine, IgE and IgG1/IgG2a ratio, thymus and activation regulated chemokine (TARC), and thymic stromal lymphopoietin (TSLP) in serum and beneficially modulated the expressions of Th1, Th2, Th17, and Th22-mediated cytokines in lesioned skin and splenocytes. Furthermore, the expressions of the skin barrier-related proteins including filaggrin (FLG), loricrin (LOR), involucrin (IVL), zonula occludens-1 (ZO-1), and occludin (OCC) were restored by F.NONI treatment. Taken together, these results suggest that F.NONI could be a therapeutic agent to attenuate AD-like skin lesions through modulating the immune balance and skin barrier function.

## 1. Introduction

Atopic dermatitis (AD) is a multifactorial, chronic inflammatory skin disorder accompanied by pruritus, dry skin, abnormal immune responses, and impaired epidermal barrier, which has no influence on normal, non-atopic individuals [1]. Various studies indicated that the complex interaction between surrounding environmental triggers and genetic mechanisms acts on triggering AD, and this is highly correlated with immune system dysregulation [2]. Worldwide, AD is one of the most common skin diseases, and its prevalence is estimated to be about 3% to 10% of adults and up to 20% of children, which has about tripled in industrialized countries over the past three decades [3,4]. Also, AD is an initial step in “atopic march” that leads to asthma, food allergy, and allergic rhinitis, which results in cutaneous diseases of systemic disorders and reduces the quality of life in afflicted patients [5].

The pathogenesis of AD is characterized by hyperactivation of a Th2-type immune response, skin barrier abnormalities, and pruritus, and the interaction of above factors brings about chronic skin inflammation [6]. Skin barrier dysfunction is one of the typical symptoms of AD and indicates defects in skin barrier proteins [7]. Compromised skin barrier integrity allows entry of external antigens and allergens, which activate immune responses across the skin surface. Under such conditions, pro-inflammatory cytokines including IL-6, IL-33, TNF-α, and thymic stromal lymphopoietin (TSLP) are produced by keratinocytes and Langerhans cells (LCs) [8]. These cytokines lead to differentiate naïve CD4^+^ T cells into Th2 cells, and then Th2 cells secrete cytokines. These cytokines secreted by Th2 cells drive the Th2 inflammatory response by immune and inflammatory cells [9,10]. The hyperactivated Th2-type immune responses induce the isotype class switch from IgM to IgE, contribute to eosinophilia, and cause pruritus [11,12]. Furthermore, increased Th2 polarization not only causes an imbalance between Th1 and Th2 cells by suppressing the function of Th1 cells in AD, but it also leads to impairing skin barrier functions by reducing epithelial barrier molecules [13]. Scratching behavior in response to pruritus caused by AD damages the skin barrier and provokes high skin sensitivity to relatively weak stimuli; as a result, subsequent pruritus exacerbates the disease cycle [6].

Corticosteroids are one of the commonly prescribed anti-inflammatory drugs for AD patients, and they primarily target T cells, which are major perpetrators of AD pathogenesis [14]. However, several studies reported that usage of corticosteroids over the long term or short term was associated with undesirable side effects such as higher rates of sepsis, dermal atrophy, rebound dermatitis, dyspigmentation, osteoporosis, venous thromboembolism, blurred vision, hypertension, and Cushing’s syndrome [15]. Calcineurin inhibitor, as nonsteroidal therapy, is approved for the treatment of AD as an alternative to corticosteroids. However, it was reported that calcineurin inhibitors may cause lymphomas, leukemias, and malignancies [16]. Thus, there is a need to develop and research safer and more effective anti-AD therapeutic products and therapies.

The in vivo model for AD using NC/Nga mice was established in 1997, and this inbred strain of mouse is widely used to investigate the pathogenesis of AD [17]. 2,4-dinitrochlorobenzene (DNCB) is extensively known as an allergenic chemical, which easily stimulates the skin surface, causing hypersensitivity of the skin and contributing to inducing dermatitis in NC/Nga mice [18]. Repeated application of DNCB induces AD-like skin lesions in mice, which indicate clinical features of human AD such as increased level of serum IgE levels and Th2 cytokines as well as a defected skin barrier [19,20].

Noni is the fruit of *Morinda citrifolia*, originating in Southeast Asia, and is distributed around Australia, the Pacific Basin, and the Caribbean [21]. Noni has been used as a beverage, fermented tonic, and folk medicine in China, Australia, Polynesia, and Hawaii as a remedy for diabetes, heart trouble, high blood pressure, skin infections, mouth sores, and toothache. It also has a variety of biological effects including anti-hypertensive, anti-bacterial, anti-inflammatory, anti-asthma, stimulation of immune system, and anti-cancer effects; therefore, it shows substantial therapeutic potentials [22]. We investigated the protective effects of fermented *Morinda citrifolia* (F.NONI) in a DNCB-induced atopic dermatitis model in vivo. This study focused not only on the AD-like skin lesion symptoms but also the immunological balance of Th1 and Th2, and skin barrier function involved in tight junction (TJ) proteins.

## 2. Materials and Methods

### 2.1. Preparation of F.NONI

The F.NONI was provided from NST Bio (Gimpo, Korea). *Morinda citrifolia* (noni) fruit was collected from the NST Bio Noni Farm Co. Ltd in French Polynesia (Indonesia islands), and F.NONI was produced in the NST bio. Briefly, harvested noni fruit was washed and frozen at −27 °C to remove bacteria. Thawed noni were sliced, incubated with 2% NST 1805 (*Lactobacillus plantarum*) probiotics and water at 30 °C for 2–3 weeks, heated at 90 °C for 30 min, and evaporated to be concentrated. After the fermentation period, the stock ferment was filtered using Whatman filter paper and vacuum filtration to eliminate debris and fruit particles from the stock solution. The obtained solution was stored at −20 °C for further use.

### 2.2. HPLC-UV Analysis of F.NONI

One gram of the F.NONI extract was diluted with 5 mL of H_2_O-MeOH (1:1) and mixed thoroughly; the solution was collected into a 5 mL volumetric flask for HPLC analysis. The extracts were combined, filtered, and then dried in a rotary evaporator under vacuum at 50 °C. The dried extracts were re-dissolved with MeOH for HPLC analysis. Chromatographic separation was performed on a Shimadzu 20A separations module coupled with 20A UV detectors, equipped with a C18 column (4.6 mm × 250 mm; 5 μm, Waters Corporation, Milford, MA, USA). The pump was connected to two mobile phases—A, MeCN; and B, 0.1% formic acid in H_2_O (v/v)—and the elute data flow rate was 0.8 mL/min. The mobile phase was consecutively programmed in linear gradients as follows: 0–5 min, 0% A; and 40 min, 30% A. The UV detector was monitored in the range of 235 nm. The injection volume was 10 µL for each of the sample solutions. The column temperature was maintained at 25 °C.

### 2.3. Animals

NC/Nga mice (*n* = 48) aged four weeks were provided by SLC (Shizuoka, Japan). The mice were kept in 55% ± 5% humidity at 23 ± 3 °C in individually ventilated cages (IVCs) under specific pathogen-free (SPF) conditions with a 12 h light–dark cycle. The mice were fed a standard laboratory diet (Central Lab Animal, Seoul, Korea) and water ad libitum. All experimental procedures were performed according to the protocol approved by the Institutional Animal Care and Use Committee guidelines of Kyung Hee University (approval no. KHUASP(SE)-18-079), and the drop-out mice were zero until the day of the final experiment.

### 2.4. Induction of AD-Like Skin Lesions and F.NONI Treatment

AD-like skin lesions were induced by DNCB (Sigma-Aldrich, St Louis, MO, USA) topical application in NC/Nga mice described in the methods of our previous study [20]. Briefly, after 1 week of acclimation, dorsal hair of NC/Nga mice was removed by using an electric shaver. After shaving hair, the mice were randomly divided into the following 6 groups, and 8 mice were allocated in each group (sample size was *n* = 8 per group): nontreated control group (Normal, naïve control group), DNCB-treated group (Control, negative control group), DNCB-treated + prednisolone 3 mg/kg (Sigma-Aldrich, St Louis, MO, USA) group (PD, positive control group), and DNCB-treated + F.NONI 250, 500, 1000 mg/kg group (F.NONI 250, F.NONI 500, F.NONI 1000). To induce AD-like skin lesions, 1% DNCB was dissolved in an acetone and ethanol mixture (2:3 v/v) and then was topically applied on the shaved dorsal area (200 µL) and right ear (100 µL) twice a week for sensitization. Following the sensitization, 0.4% DNCB dissolved in an acetone and olive oil mixture (3:1 v/v) was challenged on the dorsal skin (150 µL) and right ear (50 µL) repeatedly three times a week for 9 weeks. The mice in the normal and control groups were orally administered 0.5% carboxymethyl cellulose (0.5% CMC). Administration of PD (3 mg/kg prednisolone) and F.NONI (250, 500, 1000 mg/kg) was performed daily for 4 weeks. AD-like skin lesions were decided by dermatitis score, scratching behavior, and histological and immunological parameters.

### 2.5. Dermatitis Score and Ear Thickness

The dermatitis score was recorded three times a week as described previously (Tuesday, Thursday, and Saturday at 14:00) [23]. The scores graded as 0 (none), 1 (mild), 2 (moderate), or 3 (severe) were measured for each of the five symptoms (erythema/edema, dryness, erosion, excoriation, and lichenification). The total dermatitis score was quantified as the sum of all individual scores for five symptoms (maximum score: 15). The ear thickness was gauged on the right ear of each mice three times a week using a thickness gauge (Mitutoyo Corporation, Tokyo, Japan).

### 2.6. Scratching Behavior

The measurement of scratching behavior in experimental mice was recorded three times a week, as described in the previous study (Monday, Wednesday, and Friday at 14:00) [24]. Briefly, after vehicle administration, mice were placed in acryl cages for at least 1 h. Then, we measured and recorded the scratching movements of the neck, ears, and dorsal skin with hind paw for 30 min, which was scored from 0 to 4 (0, none; score 2, scratching shorter than 1.5 s; score 4, scratching longer than 1.5 s). The total score of scratching behavior was determined as the sum of individual measured records.

### 2.7. Histological Analysis

The dorsal skin tissue of mice was cut for histological analysis and fixed in 10% neutral formalin. Then, fixed tissues were embedded in paraffin and sliced into 4 µm thick sections. The tissue sections were stained with hematoxylin and eosin (H&E) and toluidine blue (TB). The stained tissues were supplied from the Korea pathology technical center (Cheongju, Korea). After staining, dorsal images were photographed with an optical microscope (400×, DP Controller Software; Olympus Optical, Tokyo, Japan). The thickness of skin epidermis and the number of infiltrated inflammatory cells were measured from each six locations using Image J software (National Institute of Health, Starkville, MD, USA).

### 2.8. Serum Immunoglobulin and Cytokine Analysis

Blood was immediately collected after mice were sacrificed. The method of blood collection was performed according to the protocol approved by the institute’s animal ethics committee [25]. Briefly, on the last day of the experiment, mice were anesthetized using inhalation of isoflurane (2–2.5%), and blood samples were withdrawn from venous vessels. The posterior vena cava technique was used as a terminal procedure of sacrifice. The serum samples were obtained from blood using a centrifuge (3000× *g*, 4 °C, 15 min) and stored at −80 °C until use. The levels of Ig, thymus and activation regulated chemokine (TARC), TSLP, and histamine concentrations in serum were measured using mouse enzyme-linked immunosorbent assay (ELISA) kits according to the manufacturer’s instructions. (IgE, Shibayagi, Gunma, Japan; IgG_1_ and IgG_2a_, Enzo Life Sciences, Farmingdale, NY, USA; TARC, TSLP, and histamine, Elabscience, Huston, ID, USA).

### 2.9. Splenocyte Isolation and Splenic Cytokine Analysis

Spleens were isolated from each NC/Nga mouse after the last treatment, and then examinations were performed as described [26]. Briefly, spleens were collected from NC/Nga mice aseptically, which were smashed with a sterile syringe plunger and dispersed into a single-celled suspension using a cell strainer (BD Biosciences, Franklin Lakes, NJ, USA). After treatment with red blood cell lysing buffer Hybri-Max (Sigma-Aldrich, St. Louis, MO, USA), the splenocytes were washed three times using RPMI-1640 (Gibco, Carlsbad, NY, USA) supplemented with 10% fetal bovine serum (Gemcell FBS; Gemini Bio-products, West Sacramento, CA, USA), 100 U/mL penicillin (CALSSON, Smithfield, UT, USA), and 50 mg/mL streptomycin (CALSSON, Smithfield, UT, USA). The isolated splenocytes were cultured in 24-well plates for 72 h at the concentration of 1 × 10^6^ cells/well treated with 5 μg/mL concanavalin A (Con-A) (Sigma-Aldrich, St. Louis, MO, USA) in 5% CO_2_ incubator at 37 °C. After incubation, the supernatant was collected, and splenocytes were homogenized in lysis buffer containing cOmplete™ protease inhibitor cocktail tablets (Roche Diagnostics, Indianapolis, IN, USA). The lysates were centrifuged at 10,000× *g* for 15 min at 4 °C and then collected. The collected supernatants of splenocyte and the lysates were frozen at −70 °C for subsequent cytokine analysis. The concentrations of cytokines (IL-4, IL-5, IL-6, IL-12, IL-13, IL-17, IL-22, IL-31, IL-33, and IFN-γ) in the supernatants were measured using an ELISA kit according to the manufacturer’s instructions (Elabscience, Houston, ID, USA). The concentrations of protein in the lysates were measured using the Pierce™ BCA Protein Assay Kit (Thermo Fisher Scientific, Rockford, IL, USA). The levels of cytokines in the supernatants were normalized to the protein concentrations of lysates.

### 2.10. RNA Isolation and Quantitative Real-Time Polymerase Chain Reaction (qRT-PCR) Analysis

Total RNA was obtained from dorsal skin tissues using the Easy-RED total RNA extraction kit (Intron Biotechnology, Seoul, Korea) according to the manufacturer’s instructions. After the RNA extraction, we quantified the RNA using a NanoDrop ND-1000 spectrophotometer (Thermo Fisher Scientific, Wilmington, DE, USA). Complementary DNA (cDNA) synthesis was performed using a cDNA synthesis kit (TaKaRa Bio, Shiga, Japan) with the extracted RNA. Quantitative real-time polymerase chain reaction (qRT-PCR) was executed with synthesized cDNA and SYBR Premix EX Taq (TaKaRa Bio, Shiga, Japan) into the ABI StepOnePlus™ real-time PCR system (Applied Biosystems, Waltham, MA, USA). The primer sequences are listed in Table 1. The mRNA expression level of each gene was calculated from the cycle threshold (C_t_) value using the ΔΔC_t_ method and normalized to GAPDH.

### 2.11. Western Blot Analysis

The dorsal skin tissues were smashed using a pestle after being frozen with liquid nitrogen. Subsequently, the skin tissues were homogenized with lysis buffer containing cOmplete™ protease inhibitor cocktail tablets (Roche Diagnostics, Indianapolis, ID, USA). The lysates of dorsal skin tissues were sonicated and centrifuged at 10,000× *g* for 15 min at 4 °C. The concentrations of protein from the supernatant were quantitated using the Pierce™ BCA Protein Assay Kit (Thermo Fisher Scientific, Rockford, IL, USA). After quantitation, equal amounts of protein were loaded on sodium dodecyl sulfate-polyacrylamide gel (SDS-PAGE; Bio-Rad, CA, USA) at 7.5% or 12% for electrophoresis and transferred to a polyvinylidene fluoride (PVDF) membrane. The membrane was blocked with 5% skim milk in Tris-buffered saline with 0.5% Tween-20 (TBST) and incubated at 1:1000 primary antibody overnight at 4 °C. The following day, the membranes were treated with a horseradish peroxidase-conjugated (HRP) secondary antibody (GeneTex, Inc., Irvine, CA, USA) for 2 h at a diluted concentration of 1:5000 and were visualized using a ChemiDoc™XRS + System (Bio-Rad, Richmond, CA, USA). The expression level of each protein was analyzed by Image Lab statistical software (Bio-Rad, CA, USA) and normalized to β-actin. The primary antibodies used for Western immunoblotting were as follows: filaggrin (FLG) (GeneTex, Inc., Irvine, CA, USA), loricrin (LOR) (GeneTex, Inc., Irvine, CA, USA), involucrin (IVL) (Santa Cruz, CA, USA), occluding (OCC) (Abcam, Cambridge, MA, USA), zonula occludens-1 (ZO-1) (Abcam, Cambridge, MA, USA), and β-actin (Santa Cruz, CA, USA).

### 2.12. Statistical Analysis

The data are expressed as the mean ± standard error of the mean (SEM). We performed one-way analysis of variance (ANOVA) followed by Tukey’s multiple comparison test. All analyses were performed using the Statistical Package for Social Science Software Program 23 software (SPSS; Chicago, IL, USA). Statistical significance (*p*-value) was defined as follows: ^#^
*p* < 0.05, ^##^
*p* < 0.01, and ^###^
*p* < 0.001 compared to the normal; * *p* < 0.05, ** *p* < 0.01, and *** *p* < 0.001 compared to the control.

## 3. Results

### 3.1. Identification and Quantification of Deacetylasperulosidic Acid (DAA) and Asperulosidic Acid (AA) in the F.NONI

HPLC-UV analysis at 235 nm was used to determine the components of noni and F.NONI, and the established standard chromatogram is shown in Figure 1. DAA and AA, major constituents of *Morinda citrifolia*, were found to be detected at high concentrations in noni and F.NONI. Retention time (Rt; DAA, 18.78 ± 0.06 min; AA, 27.31 ± 0.05 min) matched with the standard. Content analysis indicated that noni extracts contained 9.303 ± 0.146 mg/g of DAA and 1.470 ± 0.065 mg/g of AA. F.NONI extracts contained 13.219 ± 0.146 mg/g of DAA and 2.266 ± 0.065 mg/g of AA (Table 2).

### 3.2. F.NONI Attenuated DNCB-Induced AD-Like Skin Lesions and Scratching Behavior

The experimental procedure is summarized in Figure 2A. Repeated topical application of DNCB aggravated AD-like skin lesions, including edematous erythema, erosion, lichenification, excoriation, dryness and scratching behavior; however, the administration of F.NONI and PD indicated restoration of those lesions. Figure 2B shows representative images of dorsal skin in six groups, including Normal, Control, PD, and F.NONI 250, 500, and 1000. In Figure 2(Ca–c), the dermatitis score, ear thickness, and scratching behavior were significantly increased in the control group compared to that in the normal group. However, in Figure 2(Ca), the administration of F.NONI decreased dermatitis scores in a dose-dependent manner over time. In Figure 2(Cb), the administration of F.NONI reduced ear thickness of hyperkeratosis and dermal thickening compared with those in the control group. In addition, as shown in Figure 2(Cc), the administration of F.NONI indicated an antipruritic effect, resulting in significantly reduced scratching behavior in a dose-dependent manner compared to the control group. Administration of F.NONI 500 and 1000 tended to restore dermatitis scores, ear thickness, and scratching behavior more effectively than the PD did, which served as the positive control. Taken together, administration of F.NONI attenuated DNCB-induced AD-like skin lesions as well as scratching behavior in mice.

### 3.3. F.NONI Inhibited Thickening of the Epidermis and Infiltration of Inflammatory Cells

We observed infiltrated eosinophils, epidermal hyperplasia, and infiltrated mast cells through the H&E and TB staining in the dorsal skin. In Figure 3(Aa), H&E staining demonstrated that the thickening of the epidermis by hyperkeratosis and excoriation, and dense infiltration of eosinophils into the skin, was significantly increased in the control group compared with those in the normal group. However, the administration of F.NONI restored the epidermal thickness dose-dependently and reduced the number of eosinophils in the dorsal skin (Figure 3(B,Cb)). As shown in Figure 3(Ab,Ca), the number of mast cells in the control group also increased in dorsal skin compared to those in the normal group, whereas the administration of F.NONI decreased the number of infiltrated mast cells. Administration of F.NONI 1000 tended to decrease the infiltration of inflammatory cells and epidermis thickness more than PD did. Taken together, F.NONI administration attenuated the infiltration of inflammatory cells and restored epidermis thickness.

### 3.4. F.NONI Regulated the Levels of Immunoglobulins and Cytokines in Serum

To investigate the effect of F.NONI on AD, we measured the levels of immunoglobulins and cytokines. Th2-mediated IgE and IgG1, Th1-mediated IgG2a, and AD-related cytokine histamine, TARC, and TSLP were measured. In Figure 4A and Table 3, the levels of IgE and IgG_1_ were increased more in the control group than those in the normal group. However, IgE and IgG_1_ dose-dependently decreased in the F.NONI groups more than those in the control group, whereas the levels of IgG_2a_ were not different by F.NONI administration compared to that in the control group. Previous studies indicated that prednisolone suppressed the production of IgE, IgG_1_, and IgG_2a_ [20,27]. In our study, as an immune suppressor, the administration of PD significantly reduced the levels of IgE, IgG_1_, and IgG_2a_ compared to those in the control group. As shown in Table 3, the ratio of IgG_1_/IgG_2a_ was significantly downregulated after the administration of F.NONI in a dose-dependent manner compared to the control group. We also measured histamine, TARC, and TSLP. As shown in Figure 4C,D, the serum levels of histamine, TARC, and TSLP in the control group were increased more than those in the normal group; however, this upregulation was alleviated in the F.NONI groups in a dose-dependent manner. The efficacy of F.NONI 1000 on histamine, TARC, and TSLP tended to be similar to those in the PD. Taken together, F.NONI administration attenuated the imbalance of the immune response.

### 3.5. F.NONI Regulated the Balance of Cytokine Production in Splenocytes

To investigate the effect of F.NONI on AD-related cytokine production in the spleen, we performed an ex vivo experiment. As shown in Figure 5(Aa–e), the production of Th2-mediated cytokines IL-4, IL-5, IL-13, IL-31, and IL-33 was significantly increased in the control group, which were dose-dependently restored by the administration of F.NONI. Also, the levels of cytokines IL-6, IL-17, and IL-22 were also significantly increased in the control group more than those in the normal group, while the F.NONI group indicated a decrease greater than those in the control group (Figure 5(Ca–c)). However, the levels of Th1-mediated cytokines IL-12 and IFN-γ did not indicate a significant difference in the control group compare to that in the normal group; the administration of F.NONI tended to be increased more than that of the control group (Figure 5(Ba,b)). The administration of PD suppressed Th2-mediated cytokines as well as Th1-, Th17-, and Th22-mediated cytokines. Taken together, the administration of F.NONI modulated the balance of cytokines production in the spleen.

### 3.6. F.NONI Regulated DNCB-Induced Cytokine Gene Expression in Dorsal Skin

To investigate the effect of F.NONI on AD-related cytokine gene expression in the dorsal skin, we performed RT-qPCR analysis. As shown in Figure 6(Aa–f), the expression of Th2-mediated cytokines, including IL-4, IL-5, IL-13, IL-31, IL-33, and TSLP, were significantly increased in the control groups compared to those in the normal group. Also, the expressions of Th17, Th22, and inflammatory cytokines, including TNF-α, IL-6, IL-17A, and IL-22, were markedly upregulated in the control group (Figure 6(Ca–d)). However, the administration of F.NONI downregulated these AD-related cytokine gene expressions compare to the control group in a dose-dependent manner. On the other hand, gene expressions of Th1-mediated cytokines, including IL-12p40 and IFN-γ, were decreased in the control group more than those in the normal group, while F.NONI groups recovered these cytokine gene expressions to normal levels (Figure 6(Ba,b)). Gene expressions of all the AD-related cytokines were reduced in the PD group compared with those in the control group. Taken together, the administration of F.NONI attenuated the immune imbalance through the modulation of AD-related and inflammatory cytokine expression in dorsal skin.

### 3.7. F.NONI Restored DNCB-Induced Defects in Skin Barrier Function

To investigate the effect of F.NONI on skin barrier function, we measured skin barrier proteins and mRNA in the dorsal skin. In Figure 7(A,Ba–e), the protein levels of the skin barrier including tight junctions, such as FLG, LOR, IVL, OCC, and ZO-1, were significantly decreased in the control group compared to those in the normal group. These proteins levels of expression were dose-dependently restored in the group of F.NONI. Furthermore, in Figure 7C, the mRNA expression of pro-FLG was significantly decreased in the control group compared to that in the normal group. Decreased gene expression of pro-FLG was recovered in the F.NONI-treated groups. F.NONI 500 and 100 increased all the mRNA and protein expressions more effectively than PD. Taken together, the F.NONI treatment restored skin barrier dysfunction through the increase of mRNA and protein expression of the skin barrier, including TJs.

## 4. Discussion

Several studies have reported that AD is closely related to environmental, genetic, and immunological factors [28,29]. Immune dysregulation and skin barrier abnormalities are major features in AD, which are involved in complex pathophysiology [30]. Recent studies tried to apply the therapeutic effect of natural preparation and traditional medicine to prevent and cure AD [20,31,32]. *Morinda citrifolia* fruit, also known as noni, is reported to have antioxidant activity, which suggests that antioxidant effects may contribute to immune-modulating and anti-inflammatory activity [33]. In a previous study, noni was reported to have beneficial immunomodulatory effects in inadequate immune responses through regulating expressions of Th1 and Th2 cytokines in vivo [34]. In addition, a recent study reported that in inflammatory bowel disease, noni juice reduced expressions of inflammatory cytokines and preserved intestinal architecture [35]. Iridoids including DAA and AA are the major constituents of noni, which are known to have antioxidant and free radical scavenging activities and quench lipid peroxides [36]. Gaertneroside, a similar iridoid extracted from plants, has been shown to have regulatory effects on immune cells in vitro [37]. DAA and AA were reported to inhibit the release of TNF-α from peritoneal macrophage cells in mice [38]. We found that DAA and AA were considerably increased in the fermented noni with probiotics compared to the components of the noni in HPLC analysis. Thus, in this study, we investigated the effects of F.NONI in the AD model in vivo. The in vivo AD model using NC/Nga mice is suitable to screen potential anti-AD agents.

In our investigation, we found that the administration of F.NONI attenuated AD lesions and symptoms by DNCB. Administration of F.NONI considerably improved skin lesion severity and dose-dependently recovered clinical factors including dermatitis score, ear thickness, and scratching behavior (Figure 2). In addition, the histological analysis indicated that F.NONI treatment reduced epidermal thickness and infiltration of mast cells and eosinophils (Figure 3). Following these results, to investigate the restorative effects of F.NONI against AD, we focused on immune balance and skin barrier function through modulating expressions of AD-related cytokines.

The pathophysiology of AD has immunological abnormalities that are involved in the systemic increased Th2 response, and a combination of other T cell subset responses includes Th1, Th22, and Th17 [39,40]. Penetration of antigens and allergens through a defective epidermal barrier activates keratinocytes and secretion of TARC, TSLP, and IL-33, which acts on skin resident dendritic cells and mast cells [41,42]. Activated mast cells degranulate and secrete pro-inflammatory cytokines, including IL-4, IL-6, TNF-α, and histamine, which cause exacerbating AD-related symptoms including pruritus. Activated dendritic cells trigger polarization of Th2 cells from naïve T helper cells [43]. Differentiated Th2 cells express Th2-derived cytokines such as IL-4, IL-5, IL-13, and IL-31[44]. IL-4 and IL-13 promote isotype switching in B cells, resulting in the synthesis of IgE and generation of IgG1, and they also reduce expressions of skin barrier proteins [45]. IL-5 plays a crucial role in eosinophil development, proliferation, and survival [46]. IL-31 is an important pruritogenic inflammatory cytokine, which inhibits epidermal terminal differentiation and causes scratching behavior as well as exacerbation of skin barrier dysfunction [47,48]. The cytokines derived from Th2 re-stimulate keratinocyte and can cause increased cytokine release, including IL-6, IL-33, TNF-α, and TSLP, from keratinocyte or epidermal dendritic cells and accelerate the progression of AD [49]. In this study, the levels of histamine, TARC, and TSLP in serum were increased in the control group, while the F.NONI groups indicated the reduction of those cytokines (Figure 4B–D). Splenic productions of Th2-derived cytokines, including IL-4, IL-5, IL-13, IL-31, and IL-33, were significantly increased in the control group, whereas the administration of F.NONI and PD reduced splenic production of Th2-derived cytokines (Figure 5A). In addition, cytokine gene expressions, including IL-4, IL-5, IL-13, IL-31, IL-33, and TSLP, were increased in the dorsal skin of the control group. However, the administration of F.NONI and PD decreased the gene expressions of cytokines compared to those in the control group (Figure 6A). These results suggest that F.NONI administration attenuated upregulation of AD-related Th2 cytokines in serum, splenocytes, and dorsal skin. In addition, the decreases of Th2 cytokines such as histamine, IL-5, and IL-31 were correlated with the decreased number of infiltrated inflammatory cells and reduced scratching behavior [43,46,47,48].

The upregulation of Th2 cytokines including IL-4 is known to inhibit the generation of Th1 cells [50]. The development of Th1 cells requires the presence of IL-12 and IFN-γ [51]. IL-12 is a key factor of Th1 cell differentiation, and secreted IFN-γ by Th1 cells is known to suppress Th2 cell function and IgE synthesis as well as promote the generation of IgG_2a_ in B cells [2,52]. The serum IgG_1_/IgG_2a_ ratio is a representative marker of the Th2/Th1 balance. Increased Th2-derived cytokines and deficiency of Th1-derived cytokines are hallmarks of AD, which indicates Th2/Th1 imbalance [53]. In our study, Th1-derived cytokines including IL-12 and IFN-γ in splenocytes and dorsal skin tended to be decreased in the control group, whereas the expression of Th1-derived cytokines was normalized by the administration of F.NONI (Figure 5B and Figure 6B). Consequently, our data showed that F.NONI treatment decreased the levels of serum IgE compared to the control group (Figure 4A) and also restored the Th2/Th1 balance through regulating the serum IgG_1_/IgG_2a_ ratio (Table 3). In contrast, it was observed that the administration of PD not only inhibited expressions of Th1 and Th2 cytokines in serum, splenocytes, and skin, but it also downregulated the level of serum immunoglobulin compared to the control group. Results were already reported that prednisolone had inhibitory effects on allergic dermatitis in the mouse model [20,27]. These data indicate that the administration of F.NONI improved AD-related immune imbalance through modulating the Th1/Th2 balance in serum, splenocytes, and dorsal skin. Taken together, these results suggest that PD acted as an immune suppressor in mice, whereas the administration of F.NONI had an effect on restoring the Th1/Th2 balance through inhibiting Th2 cells and improving Th1 cells simultaneously, which regulate the level of Th1 or Th2 differentiation.

Th17 cells are correlated with the development of AD severity and produce IL-17 and IL-22 [54]. IL-17 induces the expression of various pro-inflammatory cytokines and is reported to have a crucial role in Th2 cell differentiation [55]. IL-17 also causes neutrophil recruitment leading to neutrophil- and eosinophil-mediated inflammation [56]. IL-22 is secreted by Th17 as well as Th22 and is synergistically increased with IL-17 in keratinocytes, leading to the downregulation of FLG expression [54,57]. In this study, the splenic production of IL-6, IL-17, and IL-22 was significantly increased in the control group compared with those in the normal group, while the F.NONI or PD induced a decrease in the expression of IL-6, IL-17, and IL-22 compared to those in the control group (Figure 5C). Similarly, gene expressions of IL-6, IL-17A, IL-22, and TNF-α were increased in the dorsal skin of the control group. Administration of F.NONI and PD decreased the expressions of IL-6, IL-17A, IL-22, and TNF-α compared to those in the control group (Figure 6C). These results indicate that the administration of F.NONI modulated the AD-related immune imbalance by regulating the responses of the AD-related T cell subset, including Th17 and Th22, as well as the Th1/Th2 balance.

Impaired epidermal skin is one of the hallmarks of AD. FLG is an important structural protein in the skin barrier and promotes aggregation of the keratin cytoskeleton to build up the outermost epidermal barrier [54]. In addition, FLG contributes to cell differentiation and is degraded into hydrophilic amino acids. The combination of FLG and hydrophilic amino acids is important for hydration and pH of the skin [40]. LOR and IVL, natural moisturizing factors, are also essential components of the epidermal envelope, and they are cross-linked to pro-FLG and act as reinforcement proteins in the cornified envelope [58]. Several studies reported that various cytokines were increased in AD patients and led to skin barrier dysfunction, which includes TSLP and IL-33 secreted by keratinocytes as well as Th2-derived cytokines such as IL-4, IL-13, and IL-31 [48,54,59]. In addition, IL-22 and IL-17 were also reported to downregulate the expressions of TJ proteins and terminal differentiation genes, including FLG, LOR, and IVL [39,48]. In our previous study, we reported that repeated application of DNCB on the dorsal skin of NC/Nga mice downregulated the expressions of FLG, IVL, and LOR [20]. In the present study, protein expressions of FLG, IVL, and LOR were considerably reduced in the dorsal skin of the control group, which indicated that the epidermal barrier was impaired. The administration of F.NONI significantly recovered decreased expressions of FLG, IVL, and LOR, while PD administration slightly increased the expressions of the proteins compared with the control group (Figure 7(Ba–c)).

TJs reside below the stratum corneum and regulate selective permeability into the paracellular pathway [60]. Flaky tail mice, AD model of FLG mutation, indicated skin barrier dysfunction and showed decreased expression of LOR and OCC [61]. Furthermore, several studies reported that FLG mutation in humans indicated reduced expressions of TJ proteins, including Zo-1 and OCC, and disruption of TJs results in the incorporation of LC dendrites to TJs and processing of the immune response [62,63]. The severity of AD is inversely correlated with expression levels of FLG and TJs [64]. In this study, we observed that TJ protein expressions, including ZO-1 and OCC, were also reduced in the control group compared to the normal group. However, the F.NONI dose-dependently increased the protein expressions of ZO-1 and OCC compared to those in the control group (Figure 7(Bd,e)), which indicates that F.NONI restored skin barrier impairment and dysfunction. Especially, F.NONI treatment more effectively restored TJ proteins, including Zo-1 and OCC, than PD did. These data suggest that the administration of F.NONI considerably alleviated AD skin lesions through restoring skin barrier proteins including TJs. Furthermore, these results correlated with a decreased dermatitis score and reduced expression of AD-related cytokines involved in disrupting the skin barrier in the F.NONI-treated group.

## 5. Conclusions

Administration of F.NONI significantly improved AD lesions and symptoms including dermatitis score, ear thickness, epidermal thickness, and scratching behavior. In addition, F.NONI treatment beneficially modulated the Th1/Th2 immune balance as well as Th17 and Th22 immune responses, and it reduced the infiltration of inflammatory cells. Abnormal skin barrier function was considerably restored by administration of F.NONI. Taken together, this study suggests that F.NONI could be a potential therapeutic agent to prevent and cure AD.

## Figures and Tables

**Figure 1 nutrients-12-00249-f001:**
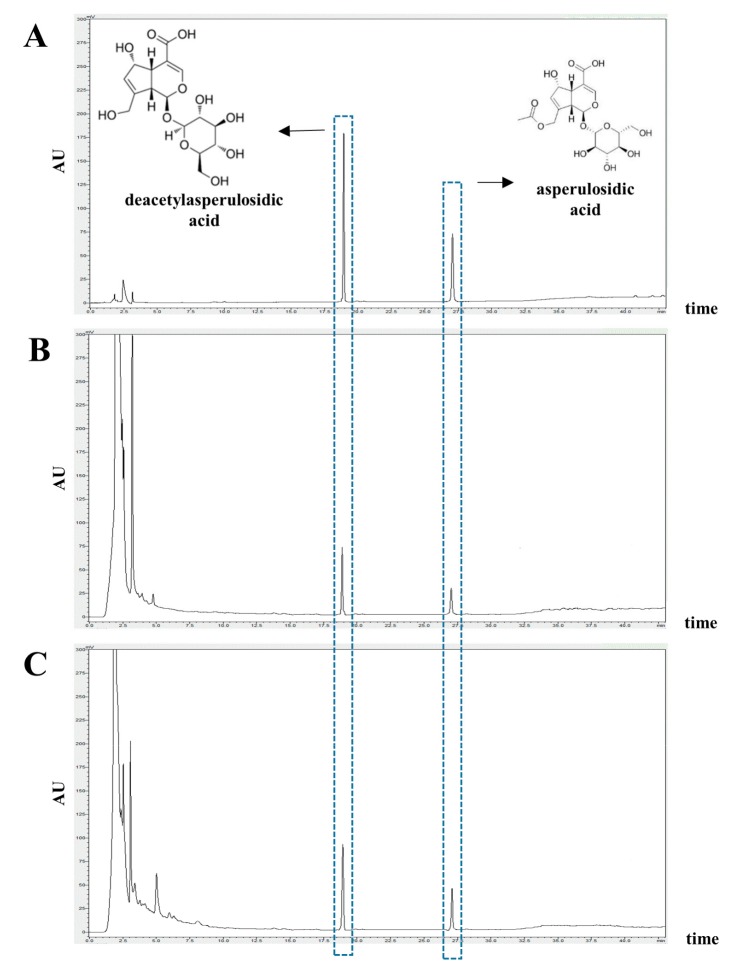
Representative HPLC-UV chromatogram of noni and F.NONI. (**A**) Standard; (**B**) Noni extract; (**C**) F.NONI extract. The arrows represent deacetylasperulosidic acid (DAA) and asperulosidic acid (AA). (Standard DDA Rt: 18.78 ± 0.06 min; noni and F.NONI each 18.78 ± 0.08 and 18.79 ± 0.07 min. Standard AA Rt: 27.31 ± 0.05 min; noni and F.NONI each 27.32 ± 0.06 and 27.32 ± 0.07 min) AU: Absorption unit.

**Figure 2 nutrients-12-00249-f002:**
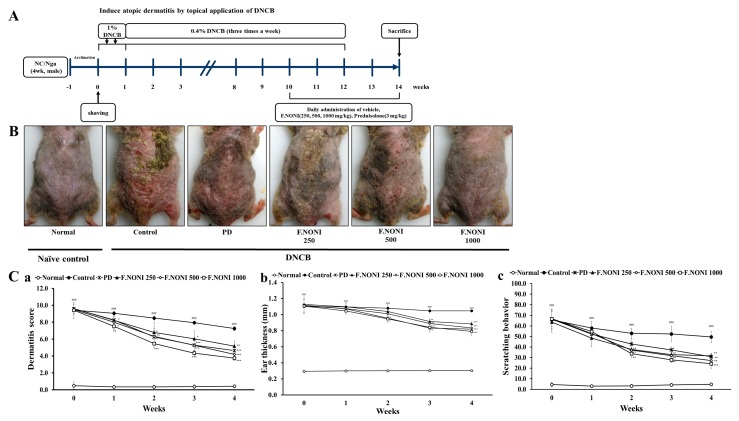
Experimental procedure and effect of F.NONI on the clinical features of AD-like symptoms induced by DNCB in NC/Nga mice. (**A**) Schematic diagram of the experimental procedure for the induction of AD lesions and F.NONI treatment. (**B**) Photographic images of skin lesions in NC/Nga mice were taken on the last day of the experiment in the fourth week. (**C**) Clinical features in NC/Nga mice—(**a**) dermatitis score, (**b**) ear thickness, and (**c**) scratching behavior—were evaluated three times a week in the term of the administration of vehicle, prednisolone, and F.NONI. The results were presented as mean ± SEM (*n* = 6). ^###^
*p* < 0.001 vs. normal, * *p* < 0.05, ** *p* < 0.01, *** *p* < 0.001 vs. control. Normal, untreated group; Control, atopic dermatitis induced by DNCB; PD, positive control (prednisolone 3 mg/kg) treated group; F.NONI (fermented *Morinda citrifolia* 250, 500, or 1000 mg/kg) treated group; DNCB (2,4-dinitrochlorobenzene).

**Figure 3 nutrients-12-00249-f003:**
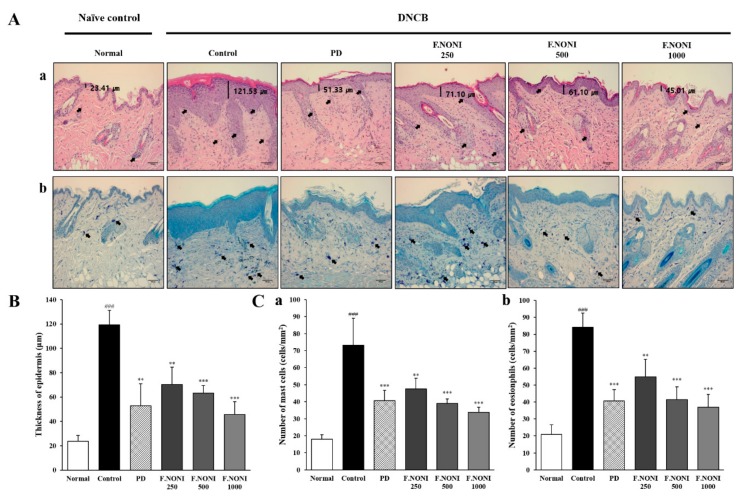
Effect of F.NONI on DNCB-induced histological features of AD-like skin lesions in NC/Nga mice. (**A**) Representative dorsal skin images of histological examination (400x, scale bar = 100μm); (**a**) H&E staining; the thicknesses of the epidermis are marked with bars, and infiltration of eosinophils is indicated by arrows; (**b**) toluidine blue staining; the infiltrations of mast cells are indicated by arrows. (**B**) The thicknesses of the epidermis were measured in the dorsal skin lesion and averaged. (**C**) The number of infiltrated inflammatory cells, (**a**) mast cells and (**b**) eosinophils, were measured in 1 mm^2^ of a dorsal skin lesion and averaged (*n* = 6). The results were presented as mean ± SEM (*n* = 6). ^###^
*p* < 0.001 vs. normal, ** *p* < 0.01, *** *p* < 0.001 vs. control. Normal, untreated group; Control, atopic dermatitis induced by DNCB; PD, positive control (prednisolone 3 mg/kg) treated group; F.NONI (fermented *Morinda citrifolia* 250, 500, or 1000 mg/kg) treated group.

**Figure 4 nutrients-12-00249-f004:**
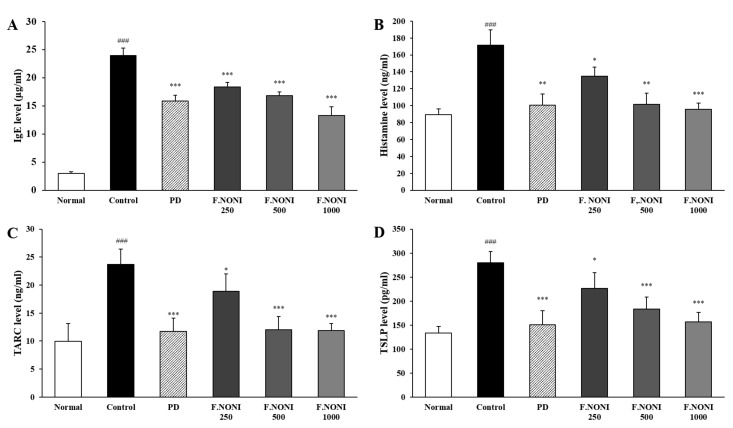
Effect of F.NONI on levels of AD-related cytokines and IgE in the serum of AD-induced NC/Nga mice. (**A**) Levels of IgE in serum; (**B**) levels of histamine in serum; (**C**) levels of TARC in serum; (**D**) levels of TSLP in serum. The sera of NC/Nga mice were collected on the last day of the experiment and measured using ELISA. The results were presented as mean ± SEM (*n* = 6). ^###^
*p* < 0.001 vs. normal, * *p* < 0.05, ** *p* < 0.01, *** *p* < 0.001 vs. control. Normal, untreated group; Control, atopic dermatitis induced by DNCB; PD, positive control (prednisolone 3 mg/kg) treated group; F.NONI (fermented *Morinda citrifolia* 250, 500, or 1000 mg/kg) treated group.

**Figure 5 nutrients-12-00249-f005:**
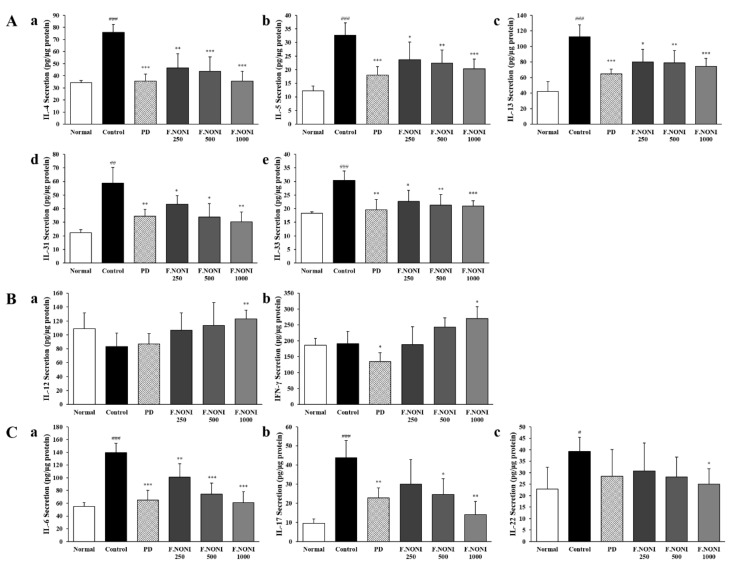
Effect of F.NONI on AD-related cytokine secretion in the splenocytes of AD-induced NC/Nga mice. (**A**) Levels of (**a**) IL-4, (**b**) IL-5, (**c**) IL-13, (**d**) IL-31, and (**e**) IL-33 in the supernatant of splenocytes. (**B**) Levels of (**a**) IL-12 and (**b**) IFN-γ in the supernatant of splenocytes. (**C**) Levels of (**a**) IL-6, (**b**) IL-17, and (**c**) IL-22 in the supernatant of splenocytes. The splenocytes of NC/Nga mice were obtained on the last day of the experiment and responded to in vitro. Con-A stimulation for 72 h, then supernatant was measured using ELISA. The results were presented as mean ± SEM (*n* = 5). ^#^
*p* < 0.05, ^##^
*p* < 0.01, ^###^
*p* < 0.001 vs. normal, * *p* < 0.05, ** *p* < 0.01, *** *p* < 0.001 vs. control. Normal, untreated group; Control, atopic dermatitis induced by DNCB; PD, positive control (prednisolone 3 mg/kg) treated group; F.NONI (fermented *Morinda citrifolia* 250, 500, or 1000 mg/kg) treated group.

**Figure 6 nutrients-12-00249-f006:**
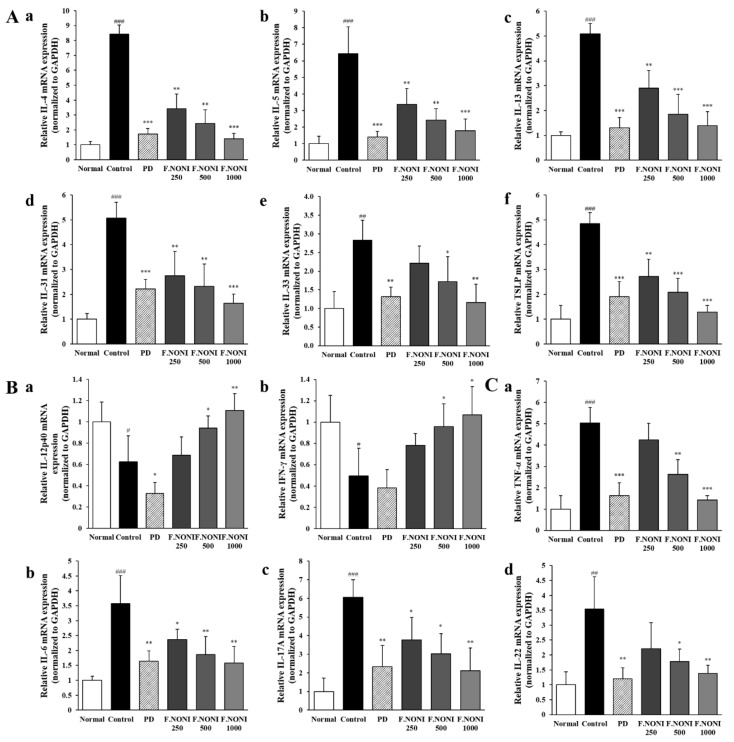
Effect of F.NONI on gene expression of AD-related cytokines in the dorsal skin of AD induced NC/Nga mice. (**A**) Gene expression of (**a**) IL-4, (**b**) IL-5, (**c**) IL-13, (**d**) IL-31, (**e**) IL-33, and (**f**) TSLP. (**B**) Gene expression of (**a**) IL-12p40 and (**b**) IFN-γ. (**C**) Gene expression of (**a**) TNF-α, (**b**) IL-6, (**c**) IL-17A, and (**d**) IL-22. The total RNA was isolated from the dorsal skins of NC/Nga mice and analyzed by RT-qPCR and then normalized to GAPDH. The results were presented as mean ± SEM (*n* = 5). ^#^
*p* < 0.05, ^##^
*p* < 0.01, ^###^
*p* < 0.001 vs. normal, * *p* < 0.05, ** *p* < 0.01, *** *p* < 0.001 vs. control. Normal, untreated group; Control, atopic dermatitis induced by DNCB; PD, positive control (prednisolone 3 mg/kg) treated group; F.NONI (fermented *Morinda citrifolia* 250, 500, or 1000 mg/kg) treated group.

**Figure 7 nutrients-12-00249-f007:**
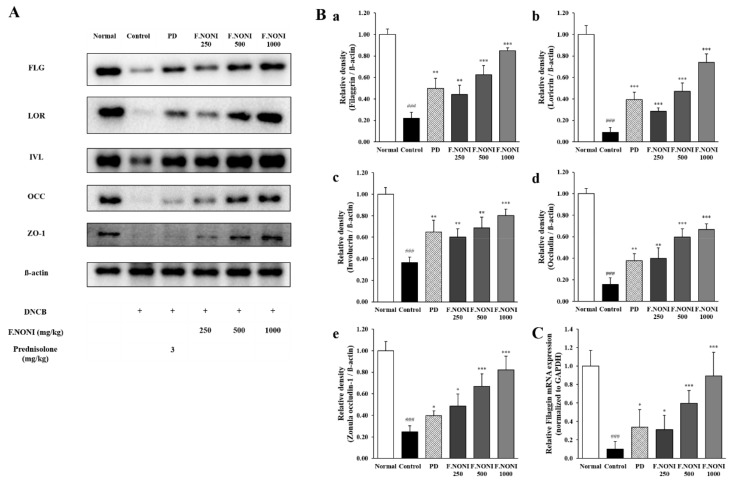
Effect of F.NONI on skin barrier dysfunction in the dorsal skin of AD-induced NC/Nga mice. (**A**) Protein levels of FLG (34 kDa), LOR (26 kDa), IVL (68 kDa), OCC (59 kDa), ZO-1 (187 kDa), and ß-actin (43 kDa) were analyzed using Western immunoblotting. (**B**) The protein levels of (**a**) FLG, (**b**) LOR, (**c**) IVL, (**d**) OCC, and (**e**) ZO-1 were quantified by band density and normalized to ß-actin. The results were presented as mean ± SEM (*n* = 4). (**C**) Gene expression levels of pro-FLG were analyzed in dorsal skin using RT-qPCR. The results were presented as mean ± SEM (*n* = 5); ^###^
*p* < 0.001 vs. normal, * *p* < 0.05, ** *p* < 0.01, *** *p* < 0.001 vs. control. Normal, untreated group; Control, atopic dermatitis induced by DNCB; PD, positive control (prednisolone 3 mg/kg) treated group; F.NONI (fermented *Morinda citrifolia* 250, 500, or 1000 mg/kg) treated group; FLG, filaggrin; LOR, loricrin; IVL, involucrin; ZO-1, zonula occludens-1; OCC, occludin; pro-FLG, pro-filaggrin.

**Table 1 nutrients-12-00249-t001:** The primer sequences (in vivo).

Gene	Sequence (Forward)	Sequence (Reverse)
*IFN-γ*	ATG AAC GCT ACA CAC TGC ATC	CCA TCC TTT TGC CAG TTC CTC
*IL-4*	ACG GGA GAA GGG ACG CCA T	GAA GCC GTA CAG ACG AGC TCA
*IL-5*	CAA AAA GAG AAG TGT GGC GAG G	TAG ATA GGA GCA GGA AGC CC
*IL-6*	TAG TCC TTC CTA CCC CAA TTT CC	TTG GTC CTT AGC CAC TCC TTC
*IL-12*	GCA GAA AGG TGC GTT CCT CG	ATG TGC AGG TGT GGT TGG GC
*IL-13*	CCT GGC TCT TGC TTG CCT T	GGT CTT GTG TGA TGT TGC TCA
*IL-17A*	AAG GCA GCA GCG ATC ATC C	GGA ACG GTT GAG GTA GTC TGA G
*IL-22*	CGA TCT CTG ATG GCT GTC CT	ACG CAA GCA TTT CTC AGA GA
*IL-31*	TCA GCA GAC GAA TCA ATA CAG C	TCG CTC AAC ACT TTG ACT TTC T
*IL-33*	TCC AAC TCC AAG ATT TCC CCG	CAT GCA GTA GAC ATG GCA GAA
*Pro-FLG*	GAA TCC ATA TTT ACA GCA AAG CAC CTT G	GGT ATG TCC AAT GTG ATT GCA CGA TTG
*TNF-α*	CAC AAG ATG CTG GGA CAG TGA	TCC TTG ATG GTG GTG CAT GA
*TSLP*	TGC AAG TAC TAG TAC GGA TGG GGC	GGA CTT CTT GTG CCA TTT CCT GAG
*GAPDH*	CGG CCG CAT CTT CTT GTG	CCG ACC TTC ACC ATT TTG TCT AC

**Table 2 nutrients-12-00249-t002:** Content of deacetylasperulosidic acid (DAA) and asperulosidic acid (AA) in the Noni and F.NONI extract (mg/g of concentrate extract).

Sample	DAA	AA
Noni	9.303 ± 0.146	1.470 ± 0.065
F.NONI	13.219 ± 0.146	2.266 ± 0.065

**Table 3 nutrients-12-00249-t003:** Effects of F.NONI on serum IgG1, IgG2a, and the IgG1/IgG2a ratio level in NC/Nga mice.

Groups	Concentration (mg/kg Body Weight)	Serum Level (μg/mL)		Normalized to Normal IgG1/IgG2a
		IgG1	IgG2a	
Normal	-	722.6 ± 107.9	597.8 ± 279.5	1.0
Control	-	2414.9 ± 215.4 ^###^	1687.8 ± 268.4 ^##^	1.8 ^##^
PD	3	1299.7 ± 317.7 ***	1038.9 ± 152.4 ***	1.2 *
F.NONI	250	1992.1 ± 251.9 *	1368.5 ± 140.3	1.7
	500	1688.7 ± 258.5 **	1389.5 ± 242.8	1.4 *
	1000	1273.6 ± 231.8 ***	1485.8 ± 116.5	1.1 **

The levels of IgG1 and IgG2a were measured in serum of NC/Nga mice using ELISA. The ratio of IgG1/IgG2a was presented in this table. The results were presented as mean ± SEM (*n* = 5). ^##^
*p* < 0.01, ^###^
*p* < 0.001 vs. normal, * *p* < 0.05, ** *p* < 0.01, *** *p* < 0.001 vs. control. Normal, untreated group; Control, only atopic dermatitis induced by DNCB; PD, positive control (prednisolone 3 mg/kg) treated group; F.NONI (fermented *Morinda citrifolia* 250, 500, or 1000 mg/kg) treated group.

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
