# Peer review of "Fermented Morinda citrifolia (Noni) Alleviates DNCB-Induced Atopic Dermatitis in NC/Nga Mice through Modulating Immune Balance and Skin Barrier Function"

_nutrients, 2020, doi:10.3390/nu12010249_

Round 1

Reviewer 1 Report

The authors well response all my concerns.

Author Response

Response to Reviewer 1 Comments

Point 1: The authors well response all my concerns. 

 Response 1: I appreciate your comment. Thanks very much.

Reviewer 2 Report

In the manuscript by Kim et al., the authors demonstrate that the extract from fermented Morinda citrifolia can reduce  

2,4-dinitrochlorobenzene (DNCB)-induced atopic dermatitis (AD)-like symptoms in NC/Nga AD model. While AD affects a substantial number of people especially Children, efficient treatment other than corticosteroids is still lacking. Thus the current study will potentially provide novel therapeutic strategies for treating AD. The experiments are well performed and conclusions are reasonable. However, afew further experiments may strengthen the current manuscript.

Based on Fig.1, the active components in F. NONI seem to be deacetylasperulosidic acid (DAA) and asperulosidic acid (AA), have the authors tried to use either DAA or AA or both in the same experimental setup? As the structure for DAA or AA has already been known, any beneficial effect from DAA or AA treatment will presumably lead to a rapid translation to therapies against AD.

Can the author clarify what “pg/ug protein” means in Fig.5?

Author Response

Response to Reviewer 2 Comments

In the manuscript by Kim et al., the authors demonstrate that the extract from fermented Morinda citrifolia can reduce 2,4-dinitrochlorobenzene (DNCB)-induced atopic dermatitis (AD)-like symptoms in NC/Nga AD model. While AD affects a substantial number of people especially Children, efficient treatment other than corticosteroids is still lacking. Thus the current study will potentially provide novel therapeutic strategies for treating AD. The experiments are well performed and conclusions are reasonable. However, a few further experiments may strengthen the current manuscript.

Point 1:  Based on Fig.1, the active components in F. NONI seem to be deacetylasperulosidic acid (DAA) and asperulosidic acid (AA), have the authors tried to use either DAA or AA or both in the same experimental setup? As the structure for DAA or AA has already been known, any beneficial effect from DAA or AA treatment will presumably lead to a rapid translation to therapies against AD.

Response 1: Thanks for your suggestion. We are performing experiments to elucidate the molecular mechanism of F.NONI therapies against AD through in vitro study using DAA and AA, and then we will prepare to submit an article. We are screening DAA and AA using immune cells including Human mast cell (HMC-1), Eosinophils (EOL), and HaCaT cell to find more effective components among DAA and AA. After screening, we will perform experiments to elucidate molecular mechanism of selected effective component in vitro study. In a previous study, DAA and AA were reported to inhibit the release of TNF-α from peritoneal macrophage cells in mouse through modulating the differentiation of helper T cell. (Cimanga,K.; Hermans, N.; Apers, et al. Complement-inhibiting Iridods from Morinda morindoides, Jornal of Natural products, 2003). Following this report, we can assume that DAA and AA had an effect on restoring AD-like skin lesion through regulating the level of Th1 or Th2 differentiation and cytokine release. We don’t know what compound will be the main effective compound now, however following our in vitro result, we will select more effective compound among DAA and AA, and then perform in vivo experiment to elucidate anti-AD effects and involved mechanism of the selected compound.

Point 2: Can the author clarify what “pg/ug protein” means in Fig.5?

Response 2: Thanks for your suggestion. The measured levels of cytokines in supernatant of splenocyte using ELISA are expressed as "pg" unit, and the measured protein concentrations of lysates using BCA protein assay are expressed as "ug protein" unit. Therefore, “pg/ug protein” means that the concentration of cytokines in supernatant of splenocytes were normalized using the protein concentrations of lysates through BCA protein assay. We added those in the Material and methods section. (2.9 splenocyte isolation and splenic cytokine analysis, line 174-183, highlighted in red color).

Briefly, Figure 5 indicates the level of AD-related cytokines in the splenocyte in NC/Nga mice. The spleen of NC/Nga mice was isolated immediately after sacrifice and smashed to single-cells. The isolated splenocytes were cultured for 72hr at the concentration of 1X106 cells/well treated with 5ug/ml Concanavalin A in CO2 incubator. After incubation, the concentration of cytokines in supernatants of splenocytes was measured using the ELISA kit. After the supernatant of splenocyte was collected, splenocyte was lysed using a lysis buffer, and then the lysates were quantitated using the BCA protein assay kit.

[line 174-183]
After incubation, the supernatant was collected and splenocytes were homogenized in lysis buffer containing cOmplete™ protease inhibitor cocktail tablets (Roche Diagnostics, Indianapolis, USA). The lysates were centrifuged at 10,000g for 15min at 4℃ and then collected. The collected supernatants of splenocyte and the lysates were frozen at -70℃ for subsequent cytokine analysis. The concentrations of cytokines (IL-4, IL-5, IL-6, IL-12, IL-13, IL-17, IL-22, IL-31, IL-33, and IFN-γ) in the supernatants were measured using ELISA kit according to the manufacturer’s instructions (Elabscience, Houston, USA). And the concentrations of protein in the lysates were measured using the Pierce™ BCA Protein Assay Kit (Thermo Fisher Scientific, Rockford, USA). The levels of cytokines in the supernatants were normalized to the protein concentrations of lysates.
